

# Measurement report: Isotopic composition of CH₄ emitted from gas exploration sites in the Transylvanian Basin, Romania

Thomas Röckmann[1], Malika Menoud[1,*], Jacoline van Es[1], Carina van der Veen[1], Hossein
Maazallahi[1,**], Pawel Jagoda[2], Jaroslav M. Necki[2], Jakub Bartyzel[2], Piotr Korben[2,3], Sara
Defratyka[2***], Martina Schmidt[3], Marius Corbu[4,5], Sebastian Iancu[4], Andreea Calcan[4,*],
Magdalena Ardelean[4], Sorin Ghemulet[4], Cristian Pop[6], Andrei Radovici[6], Alexandru
Mereuta[6], Horatiu Stefanie[6], Calin Baciu[6]

[1]Institute for Marine and Atmospheric Research Utrecht (IMAU), Utrecht University, the Netherlands
[2]Faculty of Physics and Applied Computer Science, AGH - University of Krakow, Poland
[3]Institute of Environmental Physics, University of Heidelberg, Germany
[4]National Institute for Aerospace Research "Elie Carafoli" – INCAS, Bucharest, Romania
[5]Faculty of Physics, University of Bucharest, Romania
[6]Faculty of Environmental Science and Engineering, Babes-Bolyai University, Cluj-Napoca, Romania
[*]Now at International Methane Emissions Observatory (IMEO), UNEP, Paris, France
[**]Now at Department of Renewable Energies and Environment, College of Interdisciplinary Science and
Technologies, University of Tehran, Iran
[***] Now at School of GeoSciences, University of Edinburgh, UK and National Physical Laboratory, Teddington,
UK

*Correspondence to*: Thomas Röckmann (t.rockmann@uu.nl)

**Abstract**

Isotope measurements are increasingly used to constrain the methane (CH₄) budget on various scales, from
global to regional. The success of isotope-based source attribution depends to a large degree on the knowledge
of the isotope signatures of the various source categories at the point of emission, but this information is in
many cases lacking. Here we report the isotopic composition of CH₄ emitted from 48 installations in the gas
production region of Transylvania, Romania. The isotopic source signatures are quite homogeneous across the
basin with average values of $\delta^{13}C$ = (-65.6 ± 0.5 ‰) and $\delta D$ = (-184 ± 1 ‰) confirming the biogenic origin of
the Transylvanian gas, produced by hydrogenotrophic CO₂ reduction. This is similar to values reported
previously from natural seeps in Transylvania, to the natural gas exploited in the Dolj region in Southwestern
Romania, and to the natural gas in the distribution grid in Cluj-Napoca. However, is more depleted in heavy
isotopes than the oil-associated gas emitted in the Southern Romanian Plain, and gas leakages in the city of
Bucharest. In addition, we present a step-by-step derivation of the underlying "Keeling plot" mass balance
approach that is used to derive isotope source signatures.

## 1. Introduction

Methane is a strong greenhouse gas and it is important to reduce its emissions to the atmosphere in order to reach
the goals of the Paris climate agreement (Nisbet et al., 2019;2020;Ocko et al., 2021). Emissions from the fossil
fuel sector are considered low-hanging fruit in that respect, since a large share of the emissions can be mitigated
at little or even no cost (Shindell et al., 2021;Höglund-Isaksson et al., 2020). A prerequisite for emission reduction
is knowledge of where the emissions are, which requires direct observations across the value chain. The ROMEO
project (ROmanian Methane Emissions from Oil & gas) aimed to identify, attribute and quantify emissions from
the oil and gas production infrastructure in Romania, one of the European Union's largest oil and gas production
regions. Intensive measurement campaigns with ground-based observations were carried out in 2019 in the South
Romanian Plain (Stavropoulou et al., 2023;Delre et al., 2022;Korbén et al., 2022) and in 2021 in the Transylvanian
Basin (Jagoda et al. (2025), manuscript in preparation). In addition, aircraft-borne measurements were carried out



to constrain the emissions by both in-situ measurements in 2019 (Maazallahi et al., 2024) and remote sensing in 2021 (Kuhlmann et al., 2025). Those measurements collectively demonstrated that emissions from oil and gas operations in Romania are severely underestimated in National reporting.

The isotopic composition of methane ($CH_4$) can be used to distinguish $CH_4$ that is produced via different pathways (Sherwood et al., 2017;Schwietzke et al., 2016;Menoud et al., 2021a;Quay et al., 1999;Brenninkmeijer et al., 2003;Whiticar, 2020;Milkov and Etiope, 2018;Sherwood Lollar et al., 2006;Ojeda et al., 2023). Thermogenic $CH_4$ is usually associated with relatively high $\delta^{13}C$ values between -55 and -30 ‰, and $\delta D$ generally varies in the range -250 to -100 ‰. The isotopic composition of fossil reservoirs is additionally influenced by the composition of the fuels ("dry" gas, or "wet" $CH_4$ in association with oil) and the reservoir maturity (Whiticar, 2020;Menoud et al., 2022;Milkov and Etiope, 2018). Biogenic $CH_4$ formed via the hydrogenotrophic pathway is more depleted in $^{13}C$ ($\delta^{13}C$ between -100 and -60 ‰) whereas it has relatively similar $\delta D$ values as thermogenic $CH_4$ (between -250 and -150 ‰). Biogenic $CH_4$ formed via the acetoclastic pathway has $\delta^{13}C$ values between -70 and -50 ‰ and is generally depleted in deuterium ($\delta D$ lower than -250 ‰). Abiotic $CH_4$ is relatively enriched in $^{13}C$ ($\delta^{13}C > -40$ ‰) and can cover a wide $\delta D$ range between -50 and -400 ‰. Pyrogenic $CH_4$ produced mostly during biomass burning is also enriched in both $^{13}C$ and D ($\delta^{13}C$ between -30 and -10 ‰; $\delta D > -250$ ‰) and thus falls in a similar range as abiotic $CH_4$ (Sherwood et al., 2017;Menoud et al., 2022;Whiticar, 2020).

Before the ROMEO campaigns, the existing methane isotopic data from oil and gas fields in Romania were limited to geologic natural emissions related to natural seepage. The measured values (discussed in detail below) may constitute a reference for the current work, as the mentioned sites are representative for the type of gas deposits in the study area.

Denser and systematic isotopic investigations all over Romania, combining analysis of samples collected from boreholes and from surface manifestations, are needed in order to geochemically characterize the hydrocarbon deposits, and also to better understand the methane transfer to the atmosphere from the oil and gas industry. During the ROMEO campaigns, air samples were collected in emission plumes to investigate the origin of the emitted $CH_4$ in more detail using stable isotope analysis. Menoud et al. (2022) reported the isotopic composition of samples collected at 83 ground locations and 24 samples collected on aircraft flights. They showed a wide range of isotope signatures and confirmed that the gas across the Romanian Plain is mostly associated with oil production and of thermogenic origin with average values of $\delta^{13}C = -50$ ‰ and $\delta D = -189$ ‰. A few reservoirs of microbial origin were also found. Overall, the isotope composition of gas emitted from oil and gas production sites in Romania was significantly more depleted in $^{13}C$ than commonly used values for the global fossil fuel emissions.

This study aims to provide a better isotopic characterization of $CH_4$ emissions associated with gas production in the Transylvanian Basin. We report the isotopic composition of air samples collected during phase B of the ROMEO project, conducted in summer 2021, at 48 individual gas production locations across the Transylvanian basin.

## 2. Methods

### 2.1. Campaign region

The Transylvania region is located in the central part of Romania, enclosed between the Apuseni Mountains in the West and the Eastern and Southern Carpathians. With over 100 gas fields scattered throughout the Transylvanian Basin, it remains the foremost gas producer among Central and South-Eastern European countries. The main petroleum system in the Transylvanian Basin corresponds to Neogene deposits, mainly hosting microbial methane (Popescu, 1995;Krézsek et al., 2010). Over the past century it has generated an estimated 30 TCF (trillion cubic feet) of gas (Krézsek, 2011) and undiscovered and confirmed reserves total approximately 20 TCF (Pawlewicz, 2005). There has been no discovery of commercial oil in the region of concern. Figure 1 shows the regions of the gas reservoirs, and the locations where air samples were collected for this study.

### 2.2. Sample collection and isotope measurement

When emission plumes had been identified during ground-based surveys with real-time $CH_4$ sensors, air samples were collected in the plumes by the ground teams. Air samples were pumped into 2L volume flex-foil bags using a small pump (KNF Neuberger) via Teflon tubing and a Magnesium Perchlorate dryer. Usually, two air samples were collected in the emission plume downwind of gas production installations, and several more "background samples" in clean air in the respective region, in order to determine the isotopic source signature by a Keeling plot approach (see below). A total of 96 samples from 48 individual production installations and 30 background



samples were collected across the Transylvanian Basin between June 13$^{th}$ to July 4$^{th}$, 2021. Fig. 1 shows a map of the production regions and the locations where samples were collected.

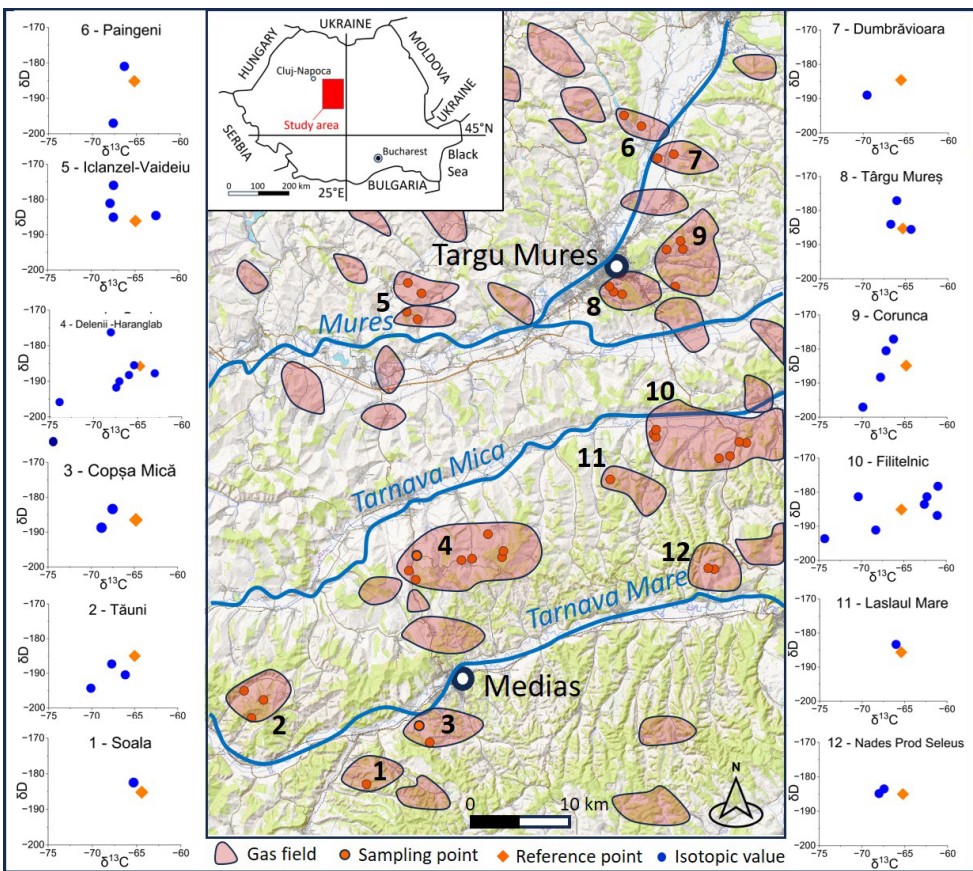

*Fig. 1: Orographic map of the campaign region with indication of the sub-surface gas fields (pink areas) and the sampling points of the samples collected for this study. The 12 small dual isotope plots on the left and right indicate the distribution of the individual source signatures derived for the 12 gas fields that were visited. The orange diamond is a common reference point ($\delta^{13}C = -65‰, \delta D = -185‰$) to help compared the values between the plots. Gas fields: 1 – Soala; 2 – Tauni; 3 – Copsa Mica; 4 – Delenii-Haranglab; 5 – Iclanzel-Vaideiu; 6 –*
*Paingeni; 7 – Dumbravioara; 8 – Targu Mures; 9 – Corunca; 10 – Filitelnic; 11 – Laslaul Mare; 12 – Nades-Prod-Seleus. (Location of the gas fields: modified after Map of the mineral resources, 1:1,000,000, Institute of Geology & Geophysics, Bucharest 1984)*

The isotopic composition of CH$_4$ ($\delta^{13}C$ and $\delta D$) in the air samples was analysed at Utrecht University using a continuous-flow isotope ratio mass spectrometry system (Brass and Röckmann, 2010;Menoud et al., 2022). First,
CH$_4$ is separated from ambient air samples and purified using temperature - controlled traps and gas chromatography. The isotopic composition of the purified CH$_4$ is then determined in an isotope ratio mass spectrometer. Individual measurements have a precision better than 0.1‰ for $\delta^{13}C$ and 2.0‰ for $\delta D$ (Brass and Röckmann, 2010;Röckmann et al., 2016;Menoud et al., 2020). The system has been carefully calibrated and participated in inter-laboratory comparisons (Brass and Röckmann, 2010;Umezawa et al., 2018). It has been used
in numerous previous projects to characterize CH$_4$ isotopic composition (Röckmann et al., 2016;Maazallahi et al., 2020;Menoud et al., 2020;2021a;2021b;Lu et al., 2021;Fernandez et al., 2022;Fiehn et al., 2023;Röckmann et al., 2011).



### 2.3. Determination of isotopic source signatures

When emissions $CH_4$ into the atmosphere lead to a clearly measurable enhancement in the $CH_4$ mole fraction, the observed mole fraction ($mf_{obs}$) is the sum of a background component ($mf_{bg}$) and a source ($mf_{src}$) component according to the mathematical equation

$$mf_{obs} = mf_{bg} + mf_{src} \qquad \text{(Eq.1)}$$

A similar equation is valid for each individual isotopologue, e.g. for the mole fraction of the $^{13}C$-substituted $CH_4$ $^{13}mf$:

$$^{13}mf_{obs} = {}^{13}mf_{bg} + {}^{13}mf_{src} \qquad \text{(Eq.2)}$$

The equation for Deuterium-substituted $CH_4$ is exactly analogous and not shown. These two mass conservation equations can be combined to determine the isotope signature of the source that is responsible for the observed mole fraction enhancement. The approach first used in (Keeling, 1961) assumes that the background component (both mole fraction and isotopic composition) remain constant over the course of the measurement. This is valid for measurements carried out over a short time close to strong emitters like the ones presented below.

Eq. 2 can be written as

$$\frac{^{13}mf_{obs}}{^{12}mf_{obs}}\,{}^{12}mf_{obs} = \frac{^{13}mf_{bg}}{^{12}mf_{bg}}\,{}^{12}mf_{bg} + \frac{^{13}mf_{src}}{^{12}mf_{src}}\,{}^{12}mf_{src}$$

$$^{13}R_{obs}\,{}^{12}mf_{obs} = {}^{13}R_{bg}\,{}^{12}mf_{bg} + {}^{13}R_{src}\,{}^{12}mf_{src} \qquad \text{(Eq.3)}$$

Where $R$ is the heavy-to light isotope ratio (in this example $^{13}R = {}^{13}C/^{12}C$). When the heavy isotope has a much lower abundance than the light isotope, the approximation $^{12}mf \sim mf$ is valid and Eq. 3 can be approximated as

$$^{13}R_{obs}\,mf_{obs} = {}^{13}R_{bg}\,mf_{bg} + {}^{13}R_{src}\,mf_{src} \qquad \text{(Eq.4)}$$

By dividing Eq. 4 by the isotope ratio of the international standard (index ST) and subtracting Eq. 1, Eq. 4 can be formulated in terms of d values as follows:

$$\frac{^{13}R_{obs}}{^{13}R_{ST}}mf_{obs} = \frac{^{13}R_{bg}}{^{13}R_{ST}}mf_{bg} + \frac{^{13}R_{src}}{^{13}R_{ST}}mf_{src}$$

$$\frac{^{13}R_{obs}}{^{13}R_{ST}}mf_{obs} - mf_{obs} = \frac{^{13}R_{bg}}{^{13}R_{ST}}mf_{bg} - mf_{bg} + \frac{^{13}R_{src}}{^{13}R_{ST}}mf_{src} - mf_{src}$$

$$\left(\frac{^{13}R_{obs}}{^{13}R_{ST}} - 1\right)mf_{obs} = \left(\frac{^{13}R_{bg}}{^{13}R_{ST}} - 1\right)mf_{bg} + \left(\frac{^{13}R_{src}}{^{13}R_{ST}} - 1\right)mf_{src}$$

$$mf_{obs} * \delta^{13}C_{obs} = mf_{bg} * \delta^{13}C_{bg} + mf_{src} * \delta^{13}C_{src} \qquad \text{(Eq.5)}$$

Noting that $mf_{bg}, \delta^{13}C_{bg}$ are assumed to be constant over the period of the measurement, Eq. 5 can be expressed as linear equation of the type $y = mx + a$ where $y = \delta^{13}C_{obs}$, $m = mf_{bg} * \left(\delta^{13}C_{bg} - \delta^{13}C_{src}\right)$ and $x = \frac{1}{mf_{obs}}$.

$$\delta^{13}C_{obs} = \frac{mf_{bg} * \left(\delta^{13}C_{bg} - \delta^{13}C_{src}\right)}{mf_{obs}} + \delta^{13}C_{src} \qquad \text{(Eq.6)}$$

The Keeling plot approach is a graphical approach where a linear fit is applied to a correlation plot of $\delta^{13}C_{obs}$ versus $\frac{1}{mf_{obs}}$, and the y-axis intercept of the linear fit equation $a$ then returns the isotopic signature of the source, $\delta^{13}C_{src}$.

We note that in cases where $mf_{bg}$ and $\delta^{13}C_{bg}$ are not constant, but can be specified (e.g. for analysis of longer time series), the mass conservation equations Eq. 1 and 2 can be rewritten differently in the so-called Miller-Tans approach (Miller and Tans, 2003) to determine isotope source signatures. The differences between the two approaches have been investigated in detail recently (Defratyka et al., 2025). That study also investigated the effect of different mathematical methods to apply linear regression analysis to Eq. 6. In our study we use the orthogonal distance regression method (Boggs et al., 1988).



## 3. Results

Fig. 2 shows a dual isotope plot of all source signatures determined at the individual sampling locations visited in this study. The locations are color-coded by gas field. Table 1 provides the numerical values, and the individual $\delta^{13}C$ values are also shown as color-coding in the map of Fig. 1. It is evident that the isotopic composition of the gas produced in the Transylvanian basin is quite homogeneous. Most of the $\delta^{13}C$ source signatures at individual sampling locations fall in a range between -70 to -60 ‰. $\delta D$ values at most individual locations fall within a narrow range between -200 and -280 ‰. This characterizes the Transylvanian gas as microbial, produced by the hydrogenotrophic pathway. All of the outliers have high uncertainties in the determination of the source signatures, implicating large scatter of individual air samples around the linear fits to the Keeling plots. Such large scatter usually indicates that the assumption of the mass balance model (Equations 1 and 2) may not be met, and in many cases this is because of other interfering sources. For example, the "high" outlier at the Dumbraviora gas field may be caused by an interference from combustion emissions.

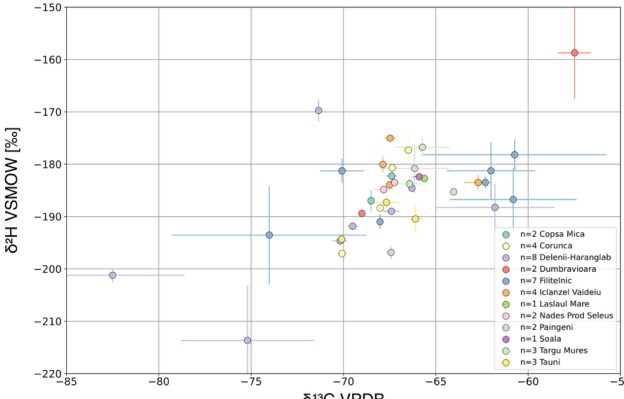

*Fig. 2: Dual isotope plot (δD versus δ¹³C) of all individual isotope source signatures derived for the different sampling locations across the Transylvanian Basin. The different symbol colours represent different gas fields.*

In order to investigate possible differences between gas fields, Fig. 3 shows the derived Keeling plot intercepts for the different gas fields, i.e., all samples from a certain gas field were combined. Fig. 3 also includes the average source signature that is derived when all of the samples are combined in one single Keeling plot analysis.

A Keeling analysis of all samples collected across the Transylvanian basin returns a y-axis intercept of $\delta^{13}C$ = (-65.6 ± 0.5 ‰) and $\delta D$ = (-184 ± 1 ‰), where the uncertainties state the 1-σ uncertainty of the intercept. This is the average isotopic composition of the $CH_4$ emitted from the investigated gas production installations across the Transylvanian basin.





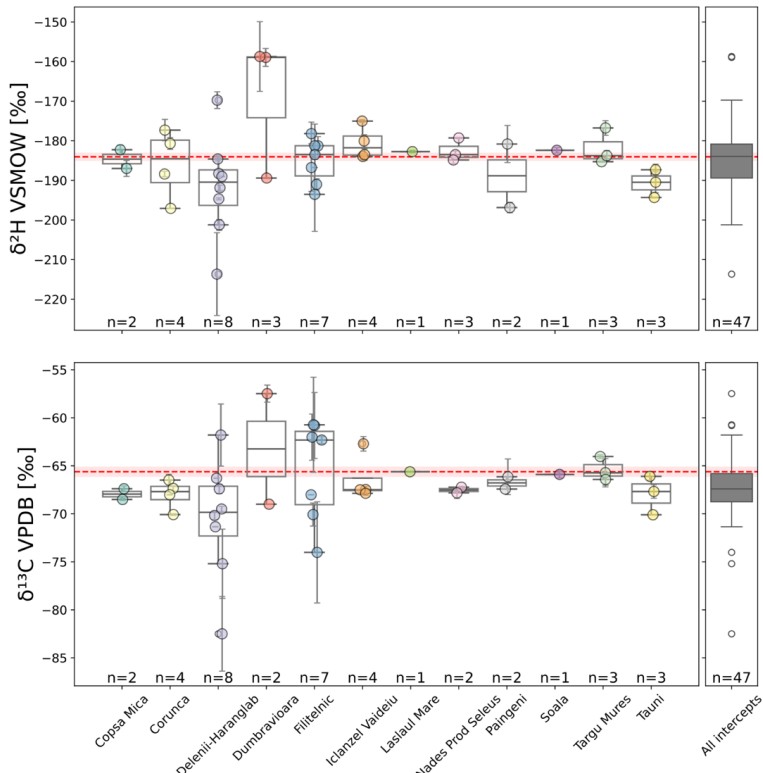

*Fig. 3: Isotope source signatures of each individual gas field visited during the ROMEO-B campaign for δD (top) and δ13C (bottom). The circles are the individual measurements, the boxes represents the 25-75 percentile of values, the mean is indicated as horizontal line, and the whiskers show the 5-95% percentiles. Note that the boxes are more an indication than a statistically robust evaluation due to the limited number of measurements. The horizontal red line shows the source signature derived from a single Keeling plot analysis combining all samples. The final category shows the distribution of all individual source signatures estimates as box plot.*

## 4.    Discussion and conclusion

Methane from several surface gas manifestations in the Transylvanian Basin has been isotopically characterized previously, showing a distinctive microbial footprint in the case of seeps for the central part of the basin, with $\delta^{13}C$ in the range of −60.3 to −67.4 ‰ and δD between −189.5 and −192.2 ‰ (Etiope, 2009;Baciu et al., 2018). The isotopic source signatures obtained from gas production infrastructure presented above are very similar to the natural gas seeps in Transylvania, suggesting that gas that escapes via natural seepage in Romania originates from similar underground reservoirs as the produced gas characterized in the present study. The formation pathway that is typically associated with these source signatures is hydrogenotrophic $CO_2$ reduction (Whiticar, 2020;Milkov and Etiope, 2018;Menoud et al., 2021a). The isotopic composition is also similar to what is found for identified gas leakages from the gas distribution network in the city of Cluj-Napoca (Es et al., 2024), confirming that the gas in the distribution grid originates from Transylvanian reservoirs. The present data allows to better define the Transylvanian reservoir in term of $CH_4$ isotopic composition in the global $CH_4$ isotopic dataset. The complete dataset is available at https://doi.org/10.18160/4SJW-ST8W (Röckmann et al., 2025).

Reported isotope values show that $CH_4$ from surface seepage becomes more thermogenic and enriched in $N_2$, $CO_2$ and He towards the eastern margin of the Transylvanian Basin, due to the thermal influence of the volcanic range of the Eastern Carpathians (Etiope et al., 2011). By contrast, the few available $CH_4$ isotopic analyses from the exterior of the Carpathian arcuated range have shown a dominant thermogenic origin of the gases. To this category



belong the mud volcanoes and everlasting fires from the Carpathian Fysch and Foredeep, as Pâclele, Fierbători, Beciu, Andreiașu (Etiope, 2009), Răiuți, Lopătari, Lepșa (Baciu et al., 2018), or the seep from Bacău – Moldavian Platform (Baciu et al., 2008).

The emitted $CH_4$ associated with oil production in the southern part of Romania has a very similar δD signature (-189 ± 38 ‰) but a very different δ¹³C signature (-49.7 ± 6.4 ‰) (Menoud et al., 2022), confirming the different (thermogenic) sub-surface formation pathway. During the ROMEO-A city campaign in the city of Bucharest, located in the southern part of the country, Fernandez et al. (2022) measured methane isotopic values of δ¹³C = −50 ‰ and δD = −196 ‰ from leakages in the natural gas distribution system in Bucharest, similar to the
associated gas in Southern Romania.

In summary, $CH_4$ emitted from 48 gas production sites in the Transylvanian basin exhibits a homogeneous isotopic composition of δ¹³C = (-65.6 ± 0.5 ‰) and δD = (-184 ± 1 ‰), confirming the biogenic origin of the gas across the basin.

## 5.    Dataset availability.

The dataset is available at https://doi.org/10.18160/4SJW-ST8W (Röckmann et al., 2025).

### Acknowledgements

This project has received funding from the European Union's Horizon 2020 Research and Innovation programme under the Marie Sklodowska-Curie grant agreement no. 722479 - MEMO², the HORIZON-CL5-2022-D1-02 program under Grant Agreement No 101081430 - PARIS and by UNEP's International Methane
Emissions Observatory (IMEO) as part of a science studies program that aims to support methane emission mitigation strategies, actions, and policies.

### Author contributions

Field campaign and sample collection: HM, PJ, JMN, JB, PK, SD, MS, MC, SI, MA, SG, CP, AR, AM, HS, CB
Isotope measurements: MM, CvdV, TR
Data analysis: JvE, MM, CvdV, SD, TR
Writing: TR, CB, JvE, SD, MS
Study design: TR, MS, JMN, AC

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
