# Peer review of "Measurement report: Isotopic composition of CH4 emitted from gas exploration sites in the Transylvanian Basin, Romania"

_EGUsphere, 2025_

## Author Comment (AC1)

**Reply to reviewer comments 1 for "Measurement report: Isotopic composition of CH$_4$ emitted from gas exploration sites in the Transylvanian Basin, Romania"**

We thank the reviewer for their constructive reviews of our manuscript and provide here a point-by-point reply. The comments from the reviewer are shown in blue and our replies in black.

This paper presents an important dataset from the Transylvanian Basin gas fields that will improve the quality of the global database on the isotopic signatures of methane from various geological settings. I recommend that it be published after some minor revisions.

The paper would benefit from additional background information, as it is currently difficult for readers unfamiliar with the Transylvanian Basin to fully understand the evolution of biogenic methane in the region. To better place the results in context, I had to conduct an independent literature search to learn about the Transylvanian Basin and gas development within the region. With a little extra work this can become a better self-contained paper.

For each of the 12 sample locations in Figure 1, it would be useful if the authors provided more detail on the regional geology, the nature of the gas traps at each site, the depth to the produced gas, and the current understanding of the local petroleum system. Such information would help readers unfamiliar with the Transylvanian Basin and make this a more valuable dataset, particularly since the gas is biogenic in origin and its formation is of significant scientific interest.

How do these data cluster according to the geological formations mapped by Tiliţă et al. (2013) (Figures 2 and 4 in that paper)? There may exist more detailed geological information, and if the authors are aware of such a source, it could replace or supplement Tiliţă et al. (2013).

**Reply:** We thank the reviewer for suggesting the improvement of the geological setting description. We have inserted the following paragraph, including more details on the specific features of the gas play.

*The study area is located in the central part of the Transylvanian Basin, a back-arc basin which is characterized by a substantial accumulation of Middle-Upper Miocene detrital sediments (Badenian to Pannonian). These deposits, formed due to fast subsidence, may exceed 5000 m in certain areas of the basin. A regressive event in the Middle Badenian provided optimal conditions for the accumulation of a substantial salt layer, potentially reaching 300 meters in thickness (Krézsek & Filipescu, 2005). The salt tectonics is responsible for the creation of brachyanticlines in the central part of the basin, and diapirs on the margins (Tiliţă et al., 2013). The commercial gas plays are mainly associated with brachyanticlines within the post-salt Badenian-Sarmatian deposits, featuring mild flank dips, typically ranging from 2 to 6°. These multi-layered structures may encompass up to 15 gas-bearing intervals, or even more in particular cases (e.g. Filitelnic – 23 pay levels) (Paraschiv, 1979). The depth of the pay intervals varies significantly, ranging from several hundred meters to over 3000 m.*

It would also be useful to see the data from this paper combined with the dataset shown in Figure 4 of Baciu et al. (2018), along with a discussion of how these results align within that graph. This addition would help readers better understand the context and evolution of the scientific discussion regarding the origin and chemical properties of the methane emitted from this unique gas field.

**Reply:** The relations between the current data and the dataset from Baciu et al. (2018) are briefly discussed in the manuscript. A more detailed work on the isotopic features of gas from Transylvania is under preparation, and will include samples collected from wells that will allow us to characterize more precisely the gas at a basin scale.

Overall, this manuscript is a valuable scientific contribution that will enhance the global isotopic methane database once these contextual revisions are addressed.

**Minor Concerns**

1. Line 23 - *"Isotope measurements are increasingly used to constrain the methane (CH4) budget on various scales, from global to regional."*

As these data are presented as being important for informing the global database, what is the estimated rate of methane emissions from the Transylvanian Basin, and what proportion does that represent of global fossil-fuel methane emissions?

**Reply:** Extensive data on the methane emission rate in the Transylvanian Basin are under preparation and will be submitted for publication soon, by the same research group.

2. Line 155 - *"For example, the "high" outlier at the Dumbraviora gas field may be caused by an interference from combustion emissions."*

What evidence supports this statement? It would be helpful if the authors provided context regarding the proximity and relative magnitude of potential combustion sources.

**Reply:** The "outlier" from Dumbravioara is contrasting with other values measured in the same gas field, and in neighbouring areas. The methodology we applied does not allow us to precisely distinguish the methane source under all circumstances. The unexpected value may result from vegetation burning or other undistinguishable source in the proximity of the sampling point.

3. *There is also a typo "Dumbraviora" -> "Dumbravioara"*

**Reply:** corrected

**Updated caption Fig. 1.**

Fig. 1. Orographic map of the campaign area, highlighting the sub-surface gas fields and the sampling locations for the samples collected in this study. The 12 small dual isotope plots on the left and right indicate the distribution of the individual source signatures derived for the 12 gas fields that were visited. The orange diamond is a common reference point for comparing the values between the plots.

**Line 180** "natural gas seeps in Transylvania" **to be replaced by** "natural gas seeps in central Transylvania"

**Line 203** "biogenic origin of the gas across the basin" **to be replaced by** "biogenic origin of the gas in the central part of the basin"

**Additional references included:**

Krézsek, C., Filipescu, S., 2005. Middle to late Miocene sequence stratigraphy of the Transylvanian Basin (Romania). Tectonophysics 410, 437–463.

Paraschiv, D., Romanian oil and gas fields. Tech. Ec. Stud., A 13, 1-382. 1979.

Tiliţă, M., Matenco, L., Dinu, C., Ionescu, L., Cloetingh, S., Understanding the kinematic evolution and genesis of a back-arc continental "sag" basin: The Neogene evolution of the Transylvanian Basin. Tectonophysics, 602, 237–258. 2013.

---

## Author Comment (AC2)

**Reply to the reviewer comment 2 for "Measurement report: Isotopic composition of CH$_4$ emitted from gas exploration sites in the Transylvanian Basin, Romania"**

We thank the reviewer for their constructive reviews of our manuscript and provide here a point-by-point reply. The comments from the reviewers are shown in blue and our replies in black.

Gas production in Romania is one of the larger methane sources in Europe (as also now shown in satellite observations). The isotopic characterisation of these emissions in this study is of interest as isotopic signatures are a good way of partitioning source types. The paper includes a good introduction to the context of this source sector and of previous studies in the region, including previous isotopic signature measurements.

A biogenic source of the methane was expected (Baciu et al., 2018). The ethane measurements on the gas in this region from previous studies should be mentioned as this also confirms a biogenic origin. Was ethane measured by the portable analysers used in this study.

**Reply:** The mobile analysers that were used during the campaign (see answer to next point) did not measure ethane online, so this cannot be confirmed.

How large did the methane mole fraction increase above background need to be to calculate the source signatures using the Keeling plot technique? This should be mentioned in the methodology section where it states that samples were collected in the emission plumes. How big were these plumes? What real-time sensors were used to find the plumes?

**Reply:** Thanks for this note, we have added the following additional description:

*We used the following real-time sensors to locate the plumes: LGR MGGA-918, Picarro G2301 or GasScouterTM G4301 (CH$_4$, CO$_2$, H$_2$O), Picarro G2203 (C$_2$H$_2$, CH$_4$). These sensors were used to quantify emission rates as presented in Jagoda et al, 2025, manuscript in preparation). We collected air samples for isotopic analysis in large plumes where the mole fraction increase was at least several ppm to hundreds of ppm, so this was never a limitation in using the Keeling technique.*

Figure 2: what do the error bars represent? They are very small in the central region, but much larger for the low and high d13C. Why is this? Please explain in the caption how the error bars were calculated.

**Reply:** They are the output of the BCES function from the BCES package in python. This function states: Bivariate correlates errors and intrinsic Scatter, translated from the FOTRAN code by Christina Bird and Matthew Bershady (Akrtis & Bershady, 1996). We have added this in the methods instead of the caption.

Line 151 – is there a typo here in the numbers for δD? Most lie between -200 and -170 ‰ (not -280).

**Reply:** Yes, this should have been -180‰, instead of -280‰.

It is interesting that the d13C isotopic signature from the gas facilities measured in this study (-65.6 ± 0.5 ‰) is much lower than the global average signature for methane from natural gas (around -44 ‰, Sherwood et al., 2017). This needs to be taken into account if incorporating isotopic measurements in regional modelling. I would have liked to have seen the implications of the findings discussed more in the discussion section.

**Reply:** This data was used in a paper interpreting continuous CH$_4$ measurements in Cluj-Napoca (van Es, 2025). The simulations in that paper indeed needed the local value for natural gas in order to reproduce the time series better. We have added a remark on this in the revised version, linked

to the suggested statement by the referee that these isotope signatures are far lower than the global average natural gas signature.